# DiffUTE: Universal Text Editing Diffusion Model

**Haoxing Chen**[1,2]**, Zhuoer Xu**[1]**, Zhangxuan Gu**[1]* **Jun Lan**[1]**, Xing Zheng**[1]**,**
**Yaohui Li**[2]**, Changhua Meng**[1]**, Huijia Zhu**[1]**, Weiqiang Wang**[1]
[1]Ant Group  [2]Nanjing University
hx.chen@hotmail.com, {xuzhuoer.xze,guzhangxuan.gzx}@antgroup.com

## Abstract

Diffusion model based language-guided image editing has achieved great success recently. However, existing state-of-the-art diffusion models struggle with rendering correct text and text style during generation. To tackle this problem, we propose a universal self-supervised text editing diffusion model (DiffUTE), which aims to replace or modify words in the source image with another one while maintaining its realistic appearance. Specifically, we build our model on a diffusion model and carefully modify the network structure to enable the model for drawing multilingual characters with the help of glyph and position information. Moreover, we design a self-supervised learning framework to leverage large amounts of web data to improve the representation ability of the model. Experimental results show that our method achieves an impressive performance and enables controllable editing on in-the-wild images with high fidelity. Our code will be avaliable in https://github.com/chenhaoxing/DiffUTE.

## 1  Introduction

Due to the significant progress of social media platforms and artificial intelligence Xu et al. [2022a], Gu et al. [2023], Zhang et al. [2022a], image editing technology has become a common demand. Specifically, AI-based technology Niu et al. [2023], Chen et al. [2023] has significantly lowered the threshold for fancy image editing, which traditionally required professional software and labor-intensive manual operations. Deep neural networks can now achieve remarkable results in various image editing tasks, such as image inpainting Feng et al. [2022], image colorization Zhang et al. [2022b], and object replacement Kwon and Ye [2022], by learning from rich paired data. Futhermore, recent advances in diffusion models Brack et al. [2023], Brooks et al. [2023], Saharia et al. [2022a] enable precise control over generation quality and diversity during the diffusion process. By incorporating a text encoder, diffusion models can be adapted to generate natural images following text instructions, making them well-suited for image editing.

Despite the impressive results, existing image editing methods still encounter numerous challenges. As a typical task, scene text editing is widely used in practical applications such as text-image synthesis, advertising photo editing, text-image correction and augmented reality translation. It aims to replace text instances (i.e., the foreground) in an image without compromising the background. However, the fine-grained and complex structures of text instances raise two major challenges: (i) **How to transfer text style and retain background texture.** Specifically, text style includes factors such as font, color, orientation, stroke size, and spatial perspective. It is difficult to precisely capture the complete text style in the source image due to the complexity of the background. (ii) **How to maintain the consistency of the edited background** especially for complex scenes, e.g., menus and street store signs.

---

*Corresponding author.

37th Conference on Neural Information Processing Systems (NeurIPS 2023).

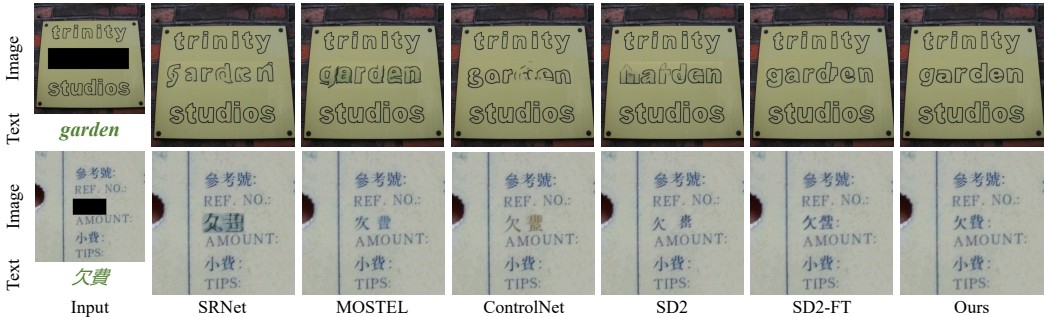

Figure 1: Examples of text editing. DiffUTE achieves the best result among existing models.

Numerous studies formulate scene text editing as a style transfer task and approach it by generative models like GANs Wu et al. [2019], Qu et al. [2023]. Typically, a cropped text region with the target style is needed as the reference image. Such methods then transfer a rendered text in the desired spelling to match the reference image's style and the source image's background. However, the two major challenges for scene text editing remains. (i) These methods are currently constrained to editing English and fail to accurately generate complex text style (e.g., Chinese). (ii) The process of cropping, transferring style and blending results in less natural-looking outcomes. End-to-end pipelines are needed for the consistency and harmony.

To address the above issues, we present DiffUTE, a general diffusion model designed to tackle high-quality multilingual text editing tasks. DiffUTE utilizes character glyphs and text locations in source images as auxiliary information to provide better control during character generation. As shown in Figure 1, our model can generate very realistic text. The generated text is intelligently matched to the most contextually appropriate text style and seamlessly integrated with the background while maintaining high quality.

The major contribution of this paper is the universal text edit diffusion model proposed to edit scene text images. DiffUTE possesses obvious advantages over existing methods in several folds:

1. We present DiffUTE, a novel universal text editing diffusion model that can edit any text in any image. DiffUTE generates high-quality text through fine-grained control of glyph and position information. DiffUTE is capable of seamlessly integrating various styles of text characters into the image context, resulting in realistic and visually pleasing outputs.

2. We design a self-supervised learning framework that enables the model to be trained with large amounts of scene text images. The framework allows the model to learn from the data without annotation, making it a highly efficient and scalable solution for scene text editing.

3. We conduct extensive experiments to evaluate the performance of DiffUTE. Our method performs favorably over prior arts for text image editing, as measured by quantitative metrics and visualization.

## 2 Preliminaries

In this paper, we adopt Stable Diffusion (SD) Rombach et al. [2022] as our baseline method to design our network architecture. SD utilizes a variational auto-encoder (VAE) to enhance computation efficiency. Through VAE, SD performs the diffusion process in low-dimensional latent space. Specifically, given an input image $x \in \mathbb{R}^{H \times W \times 3}$, the encoder $\mathcal{E}_v$ of VAE transforms it into a latent representation $z \in \mathbb{R}^{h \times w \times c}$, where $\alpha = \frac{H}{h} = \frac{W}{w}$ is the downsampling factor and $c$ is the latent feature dimension. The diffusion process is then executed in the latent space, where a conditional UNet denoiser Ronneberger et al. [2015] $\epsilon_\theta(z_t, t, y)$ is employed to predict the noise with noisy latent $z_t$, generation condition input $y$ and current time step $t$. The condition information $y$ may encompass various modalities, e.g., natural language, semantic segmentation maps and canny edge maps. To pre-processing $y$ from various modalities, SD employs a domain-specific encoder $\tau_\theta$ to project $y$ into an intermediate representation $\tau_\theta(y) \in \mathbb{R}^{M \times d_\tau}$ which is then mapped to the intermediate layers of the

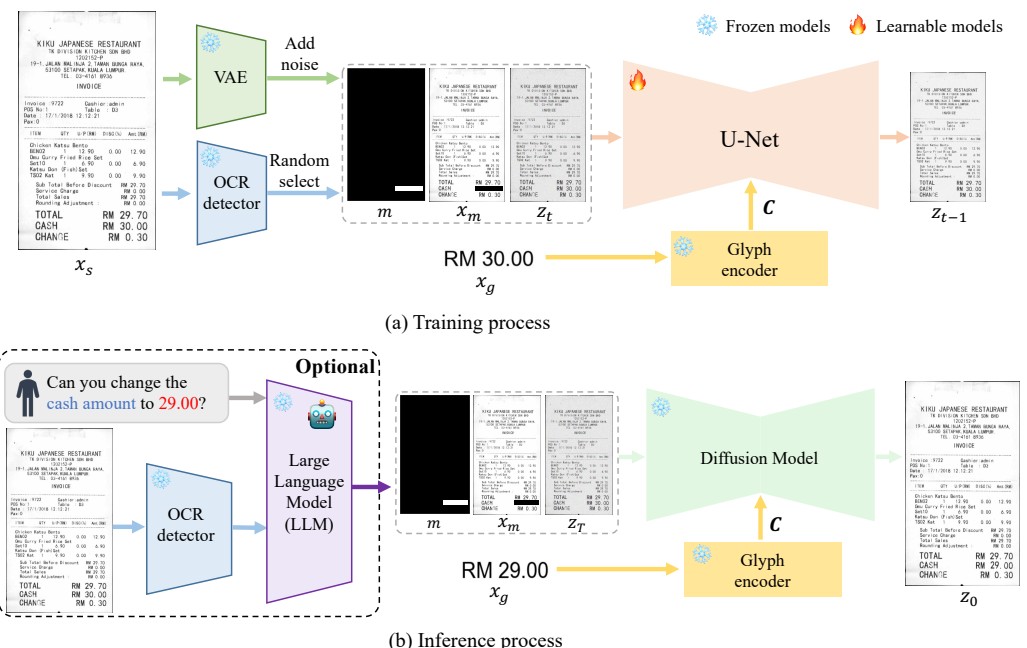

(a) Training process

(b) Inference process

Figure 2: Training and inference process of our proposed universal text editing diffusion model. (a) Given an image, we first extracted all the text and corresponding bounding boxes by the OCR detector. Then, a random area is selected and the corresponding mask and glyph image are generated. We use the embedding of the glyph image extracted by the glyph encoder as the condition, and concatenate the masked image latent vector $x_m$, mask $m$, and noisy image latent vector $z_t$ as the input of the model. (b) Users can directly input the content they want to edit, and the large language model will understand their needs and provide the areas to be edited and the target text to DiffUTE, which then completes the text editing.

UNet via a cross-attention mechanism implementing $\text{Attention}(Q, K, V) = \text{softmax}\left(\frac{QK^\top}{\sqrt{d}}\right) \cdot V$, where $Q = W_Q^{(i)} \cdot \phi_i(z_t)$, $K = W_K^{(i)} \cdot \tau_\theta(y)$, $V = W_V^{(i)} \cdot \tau_\theta(y)$. $W_Q^{(i)}, W_K^{(i)}, W_V^{(i)}$ are learnable projection matrices, $d$ denotes the output dimension of key ($K$) and query ($Q$) features, and $\phi_i(z_t) \in \mathbb{R}^{N \times d_\epsilon^i}$ denotes a flattened intermediate representation of the UNet implementing $\epsilon_\theta$. In the scenario of text-to-image generation, the condition $C = \tau_\theta(y)$ is produced by encoding the text prompts $y$ with a pre-trained CLIP text encoder $\tau_\theta$. The overall training objective of SD is defined as

$$\mathcal{L}_{sd} = \mathbb{E}_{\mathcal{E}(x), y, \epsilon \sim \mathcal{N}(0,1), t} \left[ \| \epsilon - \epsilon_\theta(z_t, t, \tau_\theta(y)) \|_2^2 \right], \tag{1}$$

Therefore, $\tau_\theta$ and $\epsilon_\theta$ can be jointly optimized via Equation (1).

## 3 Universal Text Editing Diffusion Model

### 3.1 Model Overview

The overall training process of our proposed DiffUTE method is illustrated in Figure 2 (a). Based on the cross attention mechanism in SD Rombach et al. [2022], the original input latent vector $z_t$ is replaced by the concatenation of latent image vector $z_t$, masked image latent vector $x_m$, and text mask $m$. The condition $C$ is also equipped with a glyph encoder for encoding glyph image $x_g$. Introducing text masks and glyph information enables fine-grained diffusion control throughout the training process, resulting in the improved generative performance of the model.

### 3.2 Perceptual Image Compression

Following Rombach et al. [2022], we utilize a VAE to reduce the computational complexity of diffusion models. The model learns a perceptually equivalent space to the image space but with

significantly reduced computational complexity. Since the VAE in SD is trained on natural images, its ability to restore text regions is limited. Moreover, compressing the original image directly through the VAE encoder causes the loss of dense text texture information, leading to blurry decoded images by the VAE decoder. To improve the reconstruction performance of text images, we further fine-tune the VAE on text image datasets. As shown in our experiments (Section 4.4), training VAE directly on the original image size lead to bad reconstruction results, i.e., unwanted patterns and incomplete strokes. We propose a progressive training strategy (PTT) in which the size of the images used for training increases as the training proceeds. Specifically, in the first three stages of training, we randomly crop images of sizes $S/8$, $S/4$ and $S/2$ and resize them to $S$ for training, where $S$ is the resolution of the model input image and $S = H = W$. Thus, the tuned VAE can learn different sizes of stroke details and text recovery. In the fourth stage, we train with images of the same size as the VAE input to ensure that the VAE can predict accurately when inferring.

### 3.3 Fine-grained Conditional Guidance

The pixel-level representation of text images differs greatly from the representation of natural objects. Although textual information consists of just multiple strokes of a two-dimensional structure, it has fine-grained features, and even slight movement or distortion lead to unrealistic image generation. In contrast, natural images have a much higher tolerance level as long as the semantic representation of the object is accurate. To ensure the generation of perfect text representations, we introduce two types of fine-grained guidance: positional and glyph.

**Positional guidance.** Unlike the small differences between natural images, the latent feature distributions of character pixels differ dramatically. Text generation requires attention to specific local regions instead of the existing global control conditions for natural images Zhang and Agrawala [2023], Mou et al. [2023], Cheng et al. [2023] (e.g., segmentation maps, depth maps, sketch and grayscale images). To prevent model collapse, we introduce position control to decouple the distribution between different regions and make the model focus on the region for text generation. As shown in Figure 2 (a), a binary mask is concatenated to the original image latent features.

**Glyph guidance.** Another important issue is to precisely control the generation of character strokes. Language characters are diverse and complex. For example, a Chinese character may consist of more than 20 strokes, while there are more than 10,000 common Chinese characters. Learning directly from large-scale image-text datasets without explicit knowledge guidance is complicated. Liu et al. [2022a] proposes that the character-blinded can induce robust spelling knowledge for English words only when the model parameters are larger than 100B and cannot generalize well beyond Latin scripts such as Chinese and Korean. Therefore, we heuristically incorporate explicit character images as additional conditional information to generate text accurately into the model diffusion process. As shown in Figure 2 (a), we extract the latent feature of the character image as a control condition.

### 3.4 Self-supervised Training Framework for Text Editing

It is impossible to collect and annotate large-scale paired data for text image editing, i.e., $\{(x_s, x_g, m), y\}$. It may take great expense and huge labor to manually paint reasonable editing results. Thus, we perform self-supervised training. Specifically, given an image and the OCR bounding box of a sentence in the image, our training data is composed of $\{(x_m, x_g, m), x_s\}$.

For diffusion-based inpainting models, the condition $C$ is usually text, which is usually processed by a pre-trained CLIP text encoder. Similarly, a naive solution is directly replacing it with an image encoder. To better represent glyph images, we utilize the pre-trained OCR encoder Li et al. [2023] as the glyph encoder. Such naive solution converges well on the training set. However, the generated quality is far from satisfactory for test images. We argue that the main reason is that the model learns a mundane mapping function under the naive training scheme: $x_g + x_s \cdot (1 - m) = x_s$. It impedes the network from understanding text style and layout information in the image, resulting in poor generalization. To alleviate such issue, we use a uniform font style (i.e., "arialuni") and regenerate the corresponding text image, as shown in Figure 2 (a) with the example of "RM 30.00". Thus, we prevent the model from learning such a trivial mapping function and facilitate model understanding in a self-supervised training manner.

Table 1: Quantitative comparison across four datasets. ↑ means the higher the better, underline indicates the second best method.

| Model | Web | | ArT | | TextOCR | | ICDAR13 | | Average | |
|---|---|---|---|---|---|---|---|---|---|---|
| | OCR↑ | Cor↑ | OCR↑ | Cor↑ | OCR↑ | Cor↑ | OCR↑ | Cor↑ | OCR↑ | Cor↑ |
| Pix2Pix | 17.24 | 16 | 13.52 | 11 | 15.74 | 14 | 15.48 | 15 | 15.50 | 14 |
| SRNet | 30.87 | 42 | 31.22 | 44 | 32.09 | 41 | 30.85 | 44 | 31.26 | 42.8 |
| MOSTEL | 48.93 | 61 | 60.73 | 68 | 45.97 | 53 | 53.76 | 59 | 52.35 | 60.3 |
| SD1 | 4.32 | 5 | 5.98 | 7 | 7.43 | 7 | 3.64 | 6 | 5.34 | 6.3 |
| SD2 | 5.88 | 7 | 6.94 | 9 | 9.29 | 11 | 5.32 | 8 | 6.86 | 8.8 |
| SD1-FT | 33.53 | 45 | 33.25 | 47 | 49.72 | 46 | 28.76 | 32 | 36.32 | 42.5 |
| SD2-FT | 46.34 | 51 | 49.69 | 44 | 62.89 | 59 | 46.87 | 46 | 51.45 | 50 |
| DiffSTE | 48.55 | 50 | 82.72 | 84 | 84.85 | 85 | 81.48 | 81 | 74.30 | 75 |
| DiffUTE | **84.83** | **85** | **85.98** | **87** | **87.32** | **88** | **83.49** | **82** | **85.41** | **85.5** |
| | **+35.90** | **+24** | **+3.26** | **+3** | **+2.47** | **+3** | **+2.01** | **+1** | **+11.11** | **+10.5** |

Our self-supervised training process is summarized as follows: (1) An ocr region is randomly selected from the image and the corresponding text image is regenerated with a uniform font style. (2) The regenerated character image $x_g$ is fed into glyph encoder to get condition glyph embedding $e_g$. (3) The masked image latent vector $x_m$, mask $m$ and noisy image latent vector $z_t$ is concatenated to form a new latent vector $z'_t = \text{Concat}(x_m, m, z_t)$. After dimension adjustment through a convolution layer, the feature vector $\hat{z}_t = \text{Conv}(z'_t)$ is fed into the UNet as the query component. Consequently, the training objective of DiffUTE is:

$$\mathcal{L}_{\text{DiffUTE}} = \mathbb{E}_{\mathcal{E}_v(x_s), x_g, x_m, m, \epsilon \sim \mathcal{N}(0,1), t} \left[ ||\epsilon - \epsilon_\theta(z_t, t, x_g, x_m, m)||_2^2 \right]. \tag{2}$$

### 3.5 Interactive Scene Text Editing with LLM

To enhance the interaction capability of the model, we introduced the large language model (LLM), i.e., ChatGLM Zeng et al. [2023]. Moreover, we fine-tuned ChatGLM using the extracted OCR data to facilitate a better understanding of structured information by ChatGLM, The inference process of DiffUTE is show in Figure 2 (b). We first provide the OCR information extracted by the OCR detector and the target that the user wants to edit with to LLM, which will return the target text and its corresponding bounding box. Then, we use bounding boxes to generate mask and masked images, and generate images through a complete diffusion process ($t = \{T, T - 1, ..., 0\}$) by DDIM Song et al. [2020] sampling strategy. By using ChatGLM to understand natural language instruction, we avoid requiring users to provide masks for the areas they want to edit, making our model more convenient.

## 4 Experiments

### 4.1 Data Preparation

Due to the lack of large-scale datasets for generating text image compositions, we collect 5M images by combining the web-crawled data and publicly available text image datasets, including CLDA Li, XFUND Xu et al. [2022b], PubLayNet Zhong et al. [2019] and ICDAR series competitions Zhang et al. [2019], Nayef et al. [2019], Karatzas et al. [2015], to prepare our training dataset. To verify the effectiveness of our model, we randomly selected 1000 images from ArT Chng et al. [2019], TextOCR Singh et al. [2021], ICDAR13 Karatzas et al. [2015] and web data collected by ourselves to form the test set, respectively. All the images are cropped/resized to $512 \times 512$ resolution as model inputs.

### 4.2 Implementation Details and Evaluation

**Implementation details.** Our DiffUTE consists of VAE, glyph encoder and UNet. To obtain better reconstruction ability for text images, we first fine-tuned the VAE, which is initialized from the

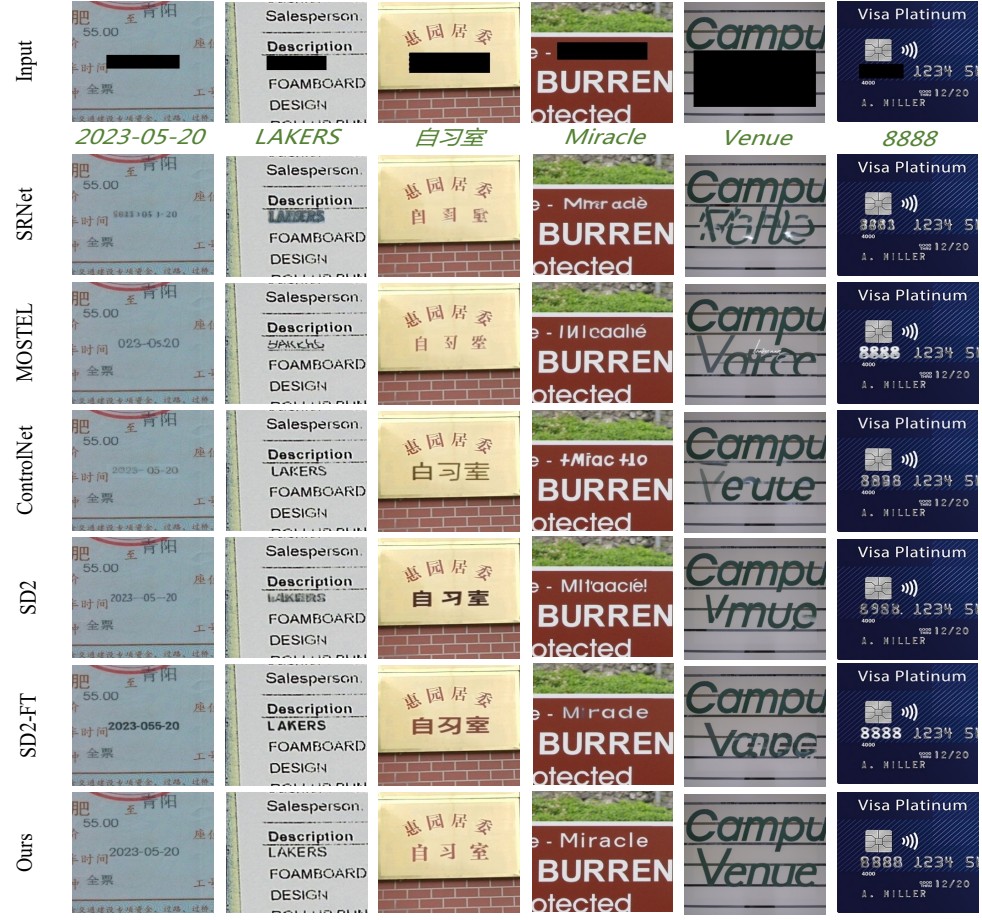

Figure 3: Visualization comparison. Our DiffUTE beats other methods with a significant improvement.

Table 2: Ablation study results. (Pos.: position control, Gly.: Glyph control.)

| Model | Web | | ArT | | TextOCR | | ICDAR13 | | Average | |
|---|---|---|---|---|---|---|---|---|---|---|
| | OCR↑ | Cor↑ | OCR↑ | Cor↑ | OCR↑ | Cor↑ | OCR↑ | Cor↑ | OCR↑ | Cor↑ |
| w/o PTT | 44.73 | 47 | 45.29 | 41 | 60.83 | 52 | 41.22 | 39 | 48.02 | 44.8 |
| w/o Pos. | 49.84 | 53 | 50.89 | 47 | 65.72 | 63 | 49.72 | 47 | 54.04 | 52.5 |
| w/o Gly. | 46.34 | 51 | 49.69 | 44 | 62.89 | 59 | 46.87 | 46 | 51.45 | 50.0 |
| DiffUTE | **84.83** | **85** | **85.98** | **87** | **87.32** | **88** | **83.49** | **82** | **85.41** | **85.5** |
| | +34.99 | +32 | +35.09 | +40 | +21.60 | +25 | +33.77 | +35 | +31.37 | +33 |

checkpoint of stable-diffusion-2-inpainting [2]. The VAE is trained for three epochs with a batch size of 48 and a learning rate of 5e-6. We use a pre-trained OCR encoder as our glyph encoder, i.e., TROCR Li et al. [2023]. During the training of DiffUTE, we set the batch size to 256, the learning rate to 1e-5, and the batch size to 5. Note that the weights of the glyph encoder and VAE were frozen during the training of DiffUTE.

**Evaluation and metrics.** In our evaluation, we evaluate the accuracy of the generated text. We report OCR accuracy, calculated separately using pre-trained recognition model Fang et al. [2021] and human evaluation of the correctness between the generated text and the target text, denoted as OCR and Cor, respectively.

---

[2] https://huggingface.co/stabilityai/stable-diffusion-2-inpainting

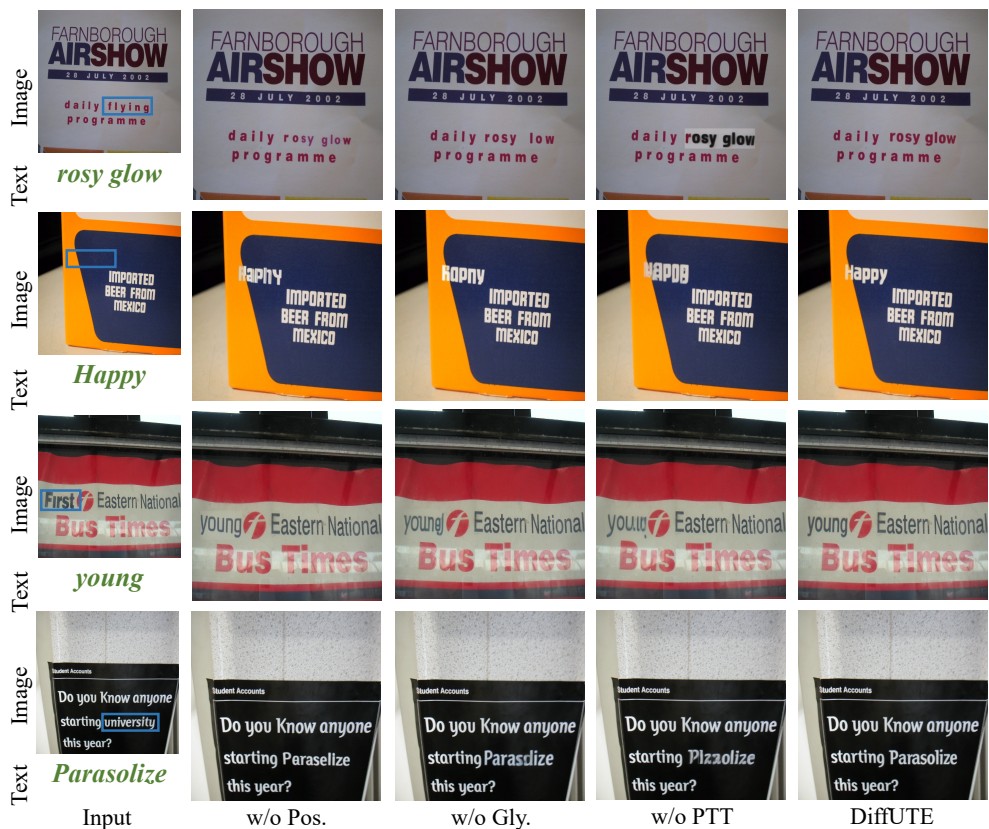

Figure 4: Sample results of ablation study.

**Baseline methods.** We compare DiffUTE with state-of-the-art scene text editing methods and diffusion models, i.e., Pix2Pix Isola et al. [2017], SRNet Wu et al. [2019], MOSTEL Qu et al. [2023], SD Rombach et al. [2022], ControlNet Zhang and Agrawala [2023] and DiffSTE Ji et al. [2023]. Pix2Pix is an image translation network. To make Pix2Pix network implement multiple style translations, we concatenate the style image and the target text in depth as the network input. Training of SRNet requires different texts to appear in the same position and background, which does not exist in real-world datasets. Therefore, we use synthtiger Yim et al. [2021] to synthesize images for fine-tuning. For MOSTEL, we fine-tuned it on our dataset. For SD, we selected two baseline methods, i.e., stable-diffusion-inpainting [3] (SD1) and stable-diffusion-2-inpainting (SD2). For fair comparison, we fine-tuned SD1 and SD2 by instruction tuning. The resulting models are termed as SD1-FT and SD2-FT. In the NLP field, instruction tuning techniques are used to train models to perform tasks based on task instructions. We aim to accurately map text instructions to corresponding text edits using the SD model. To achieve this, we constructed a dataset for fine-tuning. Each sample in the dataset consists of a language instruction describing the target text, a mask, and the ground truth. ControlNet is an image synthesis method that achieves excellent controllability by incorporating additional conditions to guide the diffusion process. To adapt this method to our text editing problem, we take the glyph image as the input to the ControlNet network. DiffSTE introduces an instruction tuning framework to train the model to learn the mapping from textual instruction to the corresponding image, and improves the pre-trained diffusion model with a dual encoder design. We followed the original setup to train DiffSTE.

### 4.3 Comparison results

The quantitative results for text generation are shown in Table 1. We can find that our DiffUTE achieves state-of-the-art results on all datasets. For example, DiffUTE improves average OCR

---

[3]https://huggingface.co/runwayml/stable-diffusion-inpainting

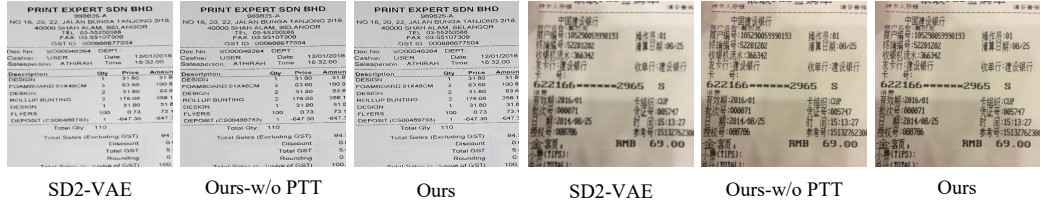

|  SD2-VAE | Ours-w/o PTT | Ours | SD2-VAE | Ours-w/o PTT | Ours |

Figure 5: Examples of image reconstruction with our method DiffUTE.

accuracy and human-evaluated text correctness by 14.95% and 14.0% compared with the second best method DiffSTE. Moreover, our method achieves better results than the diffusion model and the fine-tuned diffusion model because our fine-grained control can provide the model with prior knowledge of glyph and position. Furthermore, the poor performance of the diffusion model for instruction fine-tuning also demonstrates the superiority of our inference approach combining ChatGLM, which can achieve better editing effects.

We further conducted a visualization experiment. As shown in Figure 3, our method successfully achieved the transfer of foreground text and background texture, resulting in a regular textual structure and consistent font with the original text. Moreover, the background texture was clearer, and the overall similarity with real images was improved. In contrast, the results edited using the diffusion model often deviated from the target text, further validating the effectiveness of the glyph condition we introduced. Furthermore, other methods perform poorly when faced with more challenging Chinese text generation tasks, whereas DiffUTE still achieves good generation results.

## 4.4 Ablation results

The Ablation studies examine three main aspects, namely 1) the effectiveness of the progressive training strategy of VAE, and 2) the impact of position control and glyph control on the image generation performance of DiffUTE. The experimental results are shown in Table 2, Figure 4 and Figure 5.

**Progressive training strategy.** Without using the progressive training strategy, the editing results become distorted and the accuracy of text generation significantly decreases. The reason for such poor results is the complexity of the local structure of the text, whereby the VAE may need to learn the reconstruction ability of local details efficiently by focusing when there are too many characters in the image. And using our proposed progressive training strategy, the reconstruction ability of the model is significantly improved and more realistic results are obtained. The experimental results validate the effectiveness of this strategy and highlight the pivotal role of VAE in the diffusion model.

**Fine-grained control.** When position control is not used, the mask and masked images at the input of the UNet are removed. When glyph control is not used, the latent code obtained from the text through the CLIP text encoder is used as the condition. When position control and glyph control are not used, there is a significant drop in performance. For example, when position control is not used, the OCR accuracy of the model drops by 36.7%, and the Cor drops by 38.6%. When glyph control is not used, the model cannot generate accurate text and the OCR accuracy of the model drops by 39.8%, and the Cor drops by 41.5%. These results show that position control can help the model focus on the area where text is to be generated, while glyph control can provide prior knowledge of the shape of the characters to help the model generate text more accurately.

**Visualisation.** In Figure 6, we provide some examples edited by DiffUTE. DiffUTE consistently generates correct visual text, and the texts naturally follow the same text style, i.e. font, and color, with other surrounding texts. We can see from the experiment that DiffUTE has a strong generative power. (i) In sample N1, DiffUTE can automatically generate slanted text based on the surrounding text. (ii) As shown in sample N2, the input is 234, and DiffUTE can automatically add the decimal point according to the context, which shows that DiffUTE has some document context understanding ability. (iii) In the sample CN4, DiffUTE can generate even artistic characters very well.

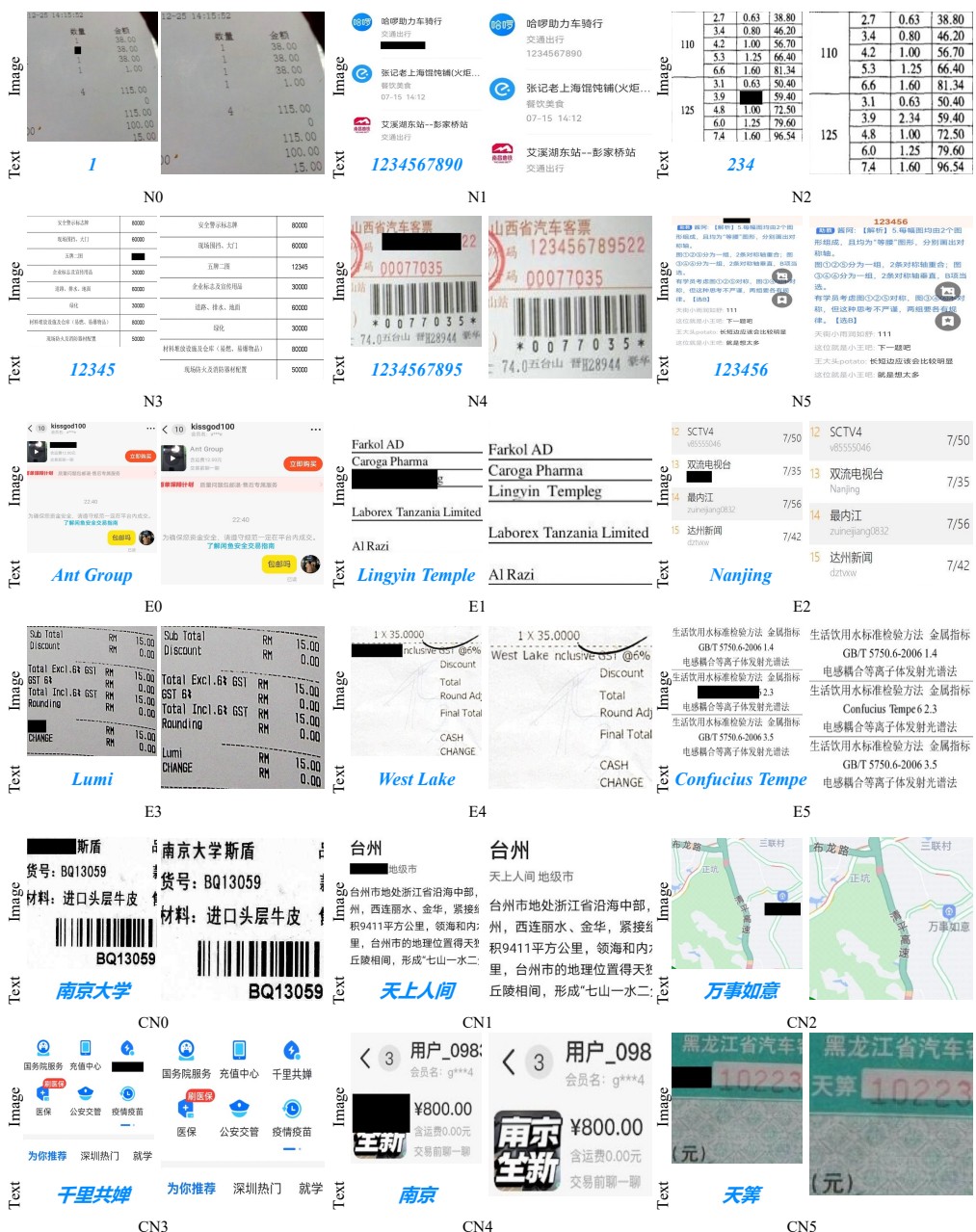

Figure 6: More visualization results of text editing.

# 5 Related Works

## 5.1 Scene Text Editing

Style transfer techniques based on Generative Adversarial Networks (GANs) have gained widespread popularity for scene text editing tasks Roy et al. [2020], Huang et al. [2022], Kong et al. [2022], Lee et al. [2021], Shimoda et al. [2021], Yang et al. [2020], Zhan et al. [2019]. These methods typically involve transferring text style from a reference image to a target text image. STEFANN Roy et al. [2020], for instance, leverages a font-adaptive neural network and a color-preserving model to edit scene text at the character level. Meanwhile, SRNet Wu et al. [2019] employs a two-step approach that involves foreground-background separation and text spatial alignment, followed by a

fusion model that generates the target text. Mostel Qu et al. [2023] improves upon these methods by incorporating stroke-level information to enhance the editing performance. However, despite their reasonable performance, these methods are often constrained in their ability to generate text in arbitrary styles and locations and can result in less natural-looking images.

## 5.2 Image Editing

Text-guided image editing has attracted increasing attention in recent years among various semantic image editing methods. Early works utilized pretrained GAN generators and text encoders to progressively optimize images based on textual prompts Bau et al. [2021], Gal et al. [2021], Pérez et al. [2003]. However, these GAN-based manipulation methods encounter difficulties in editing images with complex scenes or diverse objects, owing to the limited modeling capability of GANs. The rapid rise and development of diffusion models Rombach et al. [2022], Saharia et al. [2022b], Ruiz et al. [2023] have demonstrated powerful abilities in synthesizing high-quality and diverse images. Many studiesBrack et al. [2023], Brooks et al. [2023] have employed diffusion models for text-driven image editing. Among various diffusion models, Stable Diffusion Rombach et al. [2022] is one of the state-of-the-art models, which compresses images into low-dimensional space using an auto-encoder and implements effective text-based image generation through cross-attention layers. This model can easily adapt to various tasks, such as text-based image inpainting and image editing.

However, it has been observed that diffusion models exhibit poor visual text generation performance and are often prone to incorrect text generation. Only a few studies have focused on improving the text generation capability of diffusion models. Recently, one study trained a model to generate images containing specific text based on a large number of image-text pairs Liu et al. [2022b]. However, this work differs from ours in terms of application, as they focus on text-to-image generation, while ours concentrates on editing text in images. Another ongoing work, ControlNet Zhang and Agrawala [2023], has demonstrated remarkable performance in image editing by providing reference images such as Canny edge images and segmentation maps. While ControlNet achieves remarkably impressive results, it performs poorly on text editing tasks. To obtain better editing results, we incorporate auxiliary glyph information into the conditional generation process and emphasize local control in all diffusion steps.

## 5.3 Large Language Model

Large language models (LLMs) refer to language models that contain billions (or more) of parameters, which are trained on massive amounts of text data, such as models like GPT-3 Brown et al. [2020], Galactica Taylor et al. [2022], LLaMA Touvron et al. [2023] and ChatGLM Zeng et al. [2023]. Among them, ChatGLM is a billion-scale language model with rudimentary question-answering and conversational capabilities. It differs from BERT Devlin et al. [2018], GPT-3 and T5 Xue et al. [2021] architectures and is a self-regressive pre-training model that includes multiple objective functions. In this paper, we use ChatGLM to enhance the interaction capability of our model.

# 6 Conclusion and Limitations

In this paper, we argue that the current diffusion model can not generate realistic text in images. To tackle this problem, we present a novel method DiffUTE, a diffusion-based universal text editing model. DiffUTE generates high-quality text through fine-grained control of glyph and position information, and benefits from massive amounts of text images through a self-supervised training approach. Moreover, by integrating a large language model (i.e., ChatGLM), we can use natural language to edit the text in images, enhancing the editing usability and convenience of model. Extensive experiments have shown that DiffUTE excels in textual correctness and image naturalness.

The main limitation of our method is that the accuracy of generated text will decrease as the number of characters to be edited in the image increases. This is due to the fact that as the number of characters increase, the spatial complexity of the characters will also increase, making the generation process more challenging. Therefore, our future work will focus on improving the generation quality and solving the problem of rendering long texts.

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
