# OpenReview forum: "DiffUTE: Universal Text Editing Diffusion Model"
_NeurIPS.cc/2023/Conference — NeurIPS 2023 poster_

### Official Review · Reviewer_FkEt · 2023-06-13

**Soundness:** 2 fair
**Presentation:** 3 good
**Contribution:** 2 fair
**Rating:** 5
**Confidence:** 4

**Summary:**

This paper describes an application of diffusion models (Sohl- Dickstein et al., 2015; Ho et al., 2020) to text editing. Methodologically, this work differs from previous text diffusion (Li et al., 2022) by leveraging insights on glyph encoder and OCR detector. Empirically, this work advances the state of the art for text editing by scaling these methods to larger datasets. The paper also proposes to use self-supervised training to train the diffusion model and further explore diffusion guidance.

**Strengths:**

1. This is the work on diffusion LMs that shows results on a text editing baseline and seems to have positive results in terms of the metrics used.
2. The writing is clear and the motivations seem sound.

**Weaknesses:**

1. The main weakness is the novelty. The core idea of this paper, i.e. latent diffusion, has been demonstrated to be successful in many generation tasks. Thus it is not surprising that it works on scene text editing. Most of the techniques used in the paper have been proposed perviously.
2. The author did not provide any details regarding the position control module, thus the ablation study of this part is not convincing.
3. The authors did not evaluate a variety of evaluation measures that prior work has done such as SSIM, MSE, PSNR, and many more. These metrics should be computed to a get better idea of the quality and diversity of the output. Please see these this paper for the description of these metrics: “ Krishnan P, Kovvuri R, Pang G, et al. Textstylebrush: transfer of text aesthetics from a single example[J]. IEEE Transactions on Pattern Analysis and Machine Intelligence, 2023.”.
4. There are missing comparisions such as Krishnan et al 2023's TextStyleBrush[1] and Ji’s 2023’s DiffSTE [2].
5. The model rely on a pretrained OCR encoder which just seems like an arbitrary choice. An ablation should be provided with different pretrained encoders to understand the impact of this choice.

[1] Krishnan P, Kovvuri R, Pang G, et al. Textstylebrush: transfer of text aesthetics from a single example[J]. IEEE Transactions on Pattern Analysis and Machine Intelligence, 2023.
[2] Ji, Jiabao, et al. "Improving Diffusion Models for Scene Text Editing with Dual Encoders." arXiv preprint arXiv:2304.05568 (2023).

**Questions:**

The author mentioned that "in the first three stages of training, we randomly crop images of sizes 𝑆/8, 𝑆/4 and 𝑆/2 and resize them to 𝑆 for training", however, which are the three stages and where is the fourth one? Moreover, an ablation that would be of interest is to train with different resolutions at different stages.

**Limitations:**

Beyond the weaknesses I listed, the authors were good at addressing several limitations of this work.

---

> ### Author Rebuttal · Authors · 2023-08-04
>
> We thank the reviewer for the valuable feedback and detailed review. We would like to response as below to adress your remaining concerns.
>
> > [W1] Novelty Concern. The core idea of this paper, i.e. latent diffusion, has been demonstrated to be successful in many generation tasks. Thus it is not surprising that it works on scene text editing. Most of the techniques used in the paper have been proposed perviously.
>
> Diffusion has good performance in other tasks, but its text scene performance is unverified, and zero-shot and direct fine-tuning performance is poor ("Achilles' heel"). Achieving good results in text editing is important. DiffSTE uses instruction tuning but only supports English text editing and lacks scalability. Our multilingual text editing model prepares only corresponding images for fine-tuning new languages, due to glyph image support.
>
> > [W2] The author did not provide any details regarding the position control module, thus the ablation study of this part is not convincing.
>
> We need to clarify that we have already elaborated on the position control in Line 109-115. It is not a module, but a mask of the area to be edited. Therefore, in the ablation experiment, we compared the results without adding the mask at the input end to verify the effectiveness of the position control.
>
> > [W3] The authors did not evaluate a variety of evaluation measures that prior work has done such as SSIM, MSE, PSNR, and many more. These metrics should be computed to a get better idea of the quality and diversity of the output. Please see these this paper for the description of these metrics.
>
> We need to clarify that "diversity" is unnecessary in the scene text editing task, and the closer the generated text style is to the original text style, the better. Therefore, we conducted a user study to verify the superiority of our model in generating text styles. Specifically, we randomly select 100 images from our Web dataset. Given each image, we can obtain 4 edited results including 3 baselines and our method. We invited 50 users to identify the edited text style in each group that they felt was most similar to the original image. Finally 20,000 comparison results are collected, followed by using the Bradley-Terry (B-T) model [1] to calculate an overall ranking of all methods. As presented in the following Table, our DiffUTE achieves the highest B-T score.
>
> | Method | B-T Score |
> | :---: | :---: |
> | SRNet | 0.1140 |
> | SD2-FT | 0.1545 |
> | DiffSTE | 0.3378 |
> | DiffUTE | 0.3937 |
>
> In addition, we provide some other metrics for comparison here as well. Since Textstylebrush does not provide open source code, we do not compare with it. We compared MSE, PSNR, and SSIM on the TextOCR validation set. It should be noted that the task of natural image editing is to make the text style of the edited image similar to that of the original image and the overall comparison is harmonious. Therefore, we calculated the metrics for the entire edited image. As shown in the table below, our method achieved the best results in all metrics.
>
> | Method | MSE | PSNR | SSIM |FID |
> | :---: | :---: |  :---: |  :---: |  :---: |
> | SRNet | 0.0352 | 17.62| 0.6232 | 40.88 |
> | SD2-FT | 0.0132| 20.84 | 0.7472 | 32.52 |
> | DiffSTE | 0.0114 | 21.94 | 0.7895 | 29.84 |
> | DiffUTE | **0.0094** | **23.72** | **0.8323** |**28.22**|
>
>
> > [W4] There are missing comparisions such as Krishnan et al 2023's TextStyleBrushand Ji’s 2023’s DiffSTE.
>
> We compared our model with DiffSTE [1] on the validation set, as TextStyleBrush was not available in open-source code. The available reproduction code (https://github.com/grenlayk/text-deep-fake) differed from the original paper. As shown in the table, our method outperforms DiffSTE on all datasets, possibly due to glyph-based control conditions providing more spatial information. Also, DiffSTE only supports English text editing and struggles with more difficult Chinese text editing (Web dataset).
>
> | Model | Avg.-OCR | Avg.-Cor |Web-OCR | Web-Cor | ArT-OCR | ArT-Cor | TextOCR-OCR | TextOCR-Cor | ICDAR13-OCR | ICDAR13-Cor |
> | :---: | :---: | :---: | :---: | :---: | :---: | :---: | :---: | :---: | :---: | :---: |
> | DiffSTE | 74.30 | 75 | 48.55 | 50 | 82.72 | 84 | 84.85 | 85 | 81.48 | 81 |
> | DiffUTE | **85.41 (+11.11)** | **85.5 (+10.5)**| **84.83 (+36.28)** | **85 (+35)** | **85.98 (+3.26)** | **87 (+3)** | **87.32 (+2.47)** | **88 (+3)** | **83.49 (+2.01)** | **82 (+1)** |
>
> > [W5] The model rely on a pretrained OCR encoder which just seems like an arbitrary choice. An ablation should be provided with different pretrained encoders to understand the impact of this choice.
>
> Due to high training costs, we conducted a feasibility analysis and chose a multilingual OCR encoder, trocr, for our OCR encoder. It can support 100+ languages with image data collection and model training. Unlike DiffSTE, our model only requires OCR models to fine-tune text in different languages, which is convenient for extension.
>
> > [Q1] The author mentioned that "in the first three stages of training, we randomly crop images of sizes S/8, S/4 and S/2 and resize them to S for training", however, which are the three stages and where is the fourth one? Moreover, an ablation that would be of interest is to train with different resolutions at different stages.
>
> These trainings are specifically for VAE. As the VAE of SD is designed for natural images, its ability to restore text is weak and requires targeted fine-tuning. Our training is divided into 20 epochs, using 512-size input images and progressively increasing crop sizes (64/128/256/512) for difficulty. Starting with very large images can make VAE training difficult (with noise-like dots). Thanks for your suggestion; we'll add ablation experiments later due to high training costs.
>
> References
>
> [1] A comparative study for single image blind deblurring. In CVPR, 2016.
>
> [2] Improving Diffusion Models for Scene Text Editing with Dual Encoders. arXiv preprint arXiv:2304.05568.

---

### Official Review · Reviewer_Rgqn · 2023-07-03

**Soundness:** 3 good
**Presentation:** 3 good
**Contribution:** 2 fair
**Rating:** 6
**Confidence:** 5

**Summary:**

In this paper, the authors present DiffUTE, a universal self-supervised text editing diffusion model for language-guided image editing. They address the limitations of existing diffusion models by focusing on rendering accurate text and text style during image generation. DiffUTE incorporates modifications to the network structure, allowing it to handle multilingual character drawing using glyph and position information. Furthermore, a self-supervised learning framework leverages a large amount of web data to enhance the model's representation ability. The experimental results showcase the impressive performance of DiffUTE, demonstrating its ability to achieve high-fidelity and controllable editing on diverse real-world images. Overall, this paper presents a significant advancement in language-guided image editing and offers a promising approach for rendering realistic and customizable text in generated images.

**Strengths:**

- The problem addressed in this paper is a realistic problem that current diffusion models struggle to handle effectively.

- The incorporation of LLM  into the inference process is a compelling and intriguing approach.

**Weaknesses:**

- The paper claims significantly better results than other baselines in Table 1. However, it would be helpful to clarify if there are other baselines that have not been adequately considered.

- A simple baseline is missing. Have the authors considered directly replacing the "source text" with the "target text" and calculating the FID (Fréchet Inception Distance)?

- In Table 1, the results for SD1-FT and SD2-FT appear to be poor. It would be valuable to explain the main differences between your method and these baselines.

- There is limited mention of other methods that fine-tune the encoder-decoder. How important is this step? Additionally, could you provide details on the difference in parameter numbers shown in Table 1?

- The discussion regarding self-guidance is absent, despite the proposal of a self-supervised approach for achieving text editing in diffusion. The related papers are:

- Self-Guided Diffusion Models
- Why Are Conditional Generative Models Better Than Unconditional Ones?
- Visual Chain-of-Thought Diffusion Models

**Questions:**

As Above

**Limitations:**

Yes

---

> ### Author Rebuttal · Authors · 2023-08-04
>
> We thank the reviewer for the valuable feedback and detailed review. We would like to response as below to adress your remaining concerns.
>
> > [W1] The paper claims significantly better results than other baselines in Table 1. However, it would be helpful to clarify if there are other baselines that have not been adequately considered.
>
> Thank you for your professional comments. According to the opinions of other reviewers, we have added a comparison with the latest DiffSTE[1]. As shown in the table below, our method performs better than DiffSTE on all datasets, which may be due to the use of instructions to control image editing in DiffSTE. Obviously, glyph-based control conditions can provide more spatial information. In addition, DiffSTE only supports English text editing and does not perform well on more difficult Chinese text editing.
>
> | Model | Avg.-OCR | Avg.-Cor |Web-OCR | Web-Cor | ArT-OCR | ArT-Cor | TextOCR-OCR | TextOCR-Cor | ICDAR13-OCR | ICDAR13-Cor |
> | :---: | :---: | :---: | :---: | :---: | :---: | :---: | :---: | :---: | :---: | :---: |
> | DiffSTE | 74.30 | 75 | 48.55 | 50 | 82.72 | 84 | 84.85 | 85 | 81.48 | 81 |
> | DiffUTE | **85.41 (+11.11)** | **85.5 (+10.5)**| **84.83 (+36.28)** | **85 (+35)** | **85.98 (+3.26)** | **87 (+3)** | **87.32 (+2.47)** | **88 (+3)** | **83.49 (+2.01)** | **82 (+1)** |
>
> > [W2] A simple baseline is missing. Have the authors considered directly replacing the "source text" with the "target text" and calculating the FID?
>
> Thank you for your suggestion. We added a simple method as a baseline, as shown in the table below, and compared different methods in terms of FID. Specifically, our baseline is divided into two steps. In the first step, we use traditional inpainting algorithms[2] to restore the areas that need to be tampered with, and then directly write the desired text in those areas.
>
> | Method | MSE | PSNR | SSIM |FID |
> | :---: | :---: |  :---: |  :---: |  :---: |
> | Baseline | 0.0894 | 11.04 | 0.3523 | 90.82 |
> | SRNet | 0.0352 | 17.62| 0.6232 | 40.88 |
> | SD2-FT | 0.0132| 20.84 | 0.7472 | 32.52 |
> | DiffSTE | 0.0114 | 21.94 | 0.7895 | 29.84 |
> | DiffUTE | **0.0094** | **23.72** | **0.8323** |**28.22**|
>
> > [W3] In Table 1, the results for SD1-FT and SD2-FT appear to be poor. It would be valuable to explain the main differences between your method and these baselines.
>
> The reason why SD1-FT and SD2-FT perform poorly is because they perform editing through instruction. It is difficult for instruction to provide information about the shape of the text, and there are also errors in understanding the specific text to be edited from the instruction. Our method uses glyph image to provide information about the shape of the text, and adds position control to strengthen the generation target, which provides richer control information than SD and generates high-quality text. In addition, since we use glyph image as the condition, when facing the editing task of a new language, only the image of the text in the new language needs to be provided for fine-tuning our model. In contrast, SD needs to prepare corresponding instructions for fine-tuning.
>
> > [W4] There is limited mention of other methods that fine-tune the encoder-decoder. How important is this step? Additionally, could you provide details on the difference in parameter numbers shown in Table 1?
>
> In our model, there are two fine-tuning processes. In fine-tuning the VAE, there are two options: training directly using the input image size of the model, and training using the progressive training strategy. We compare these two methods in Figure 5 and Figure 6. In fine-tuning the entire SD, there are no other options for fine-tuning methods. I have listed the detailed parameters of the models in Table 1 below. Unfortunately, many models did not provide parameter information, which may be because the editing effect in this field is currently not satisfactory enough. When the effect becomes good enough, people will consider lightweight.
>
>
> | Method | Params |
> | :---: | :---: |
> | Pix2Pix | Unknown|
> | SRNet | Unknown |
> | MOSTEL | Unknow |
> | SD| 1070 MB |
> | DiffSTE | Unknown |
> | DiffUTE | 1500 MB |
>
>
> > [W5] The discussion regarding self-guidance is absent, despite the proposal of a self-supervised approach for achieving text editing in diffusion. The related papers are: Self-Guided Diffusion Models; Why Are Conditional Generative Models Better Than Unconditional Ones?;Visual Chain-of-Thought Diffusion Models.
>
> Thank you for your professional comments. We will add the following discussion about other self-supervised diffusion models in the revised paper.
>
> Hu et al. proposed self-guided diffusion models, which use the flexibility of self-supervised signals to design the framework of self-guided diffusion models to eliminate the need for annotations. By leveraging a feature extraction function and a selfannotation function, they provides guidance signals at various image granularities: from the level of holistic images to object boxes and even segmentation masks. Bao et al. train a conditional diffusion model by taking the cluster indices as conditions. And Harvey et al. propose to close the gap between conditional and unconditional models using a two-stage sampling procedure. The above methods mostly train by obtaining pseudo-labels of the image, while DiffUTE is essentially a task similar to MAE. It is hoped that in this process, the model can learn the mapping relationship between glyph, surrounding background, and image representation.
>
> References
>
> [1] Improving Diffusion Models for Scene Text Editing with Dual Encoders. arXiv preprint arXiv:2304.05568.
>
> [2] Navier-stokes, fluid dynamics, and image and video inpainting. In CVPR, 2001.
>
> [3] Self-Guided Diffusion Models. arXiv preprint arXiv:2210.06462.
>
> [4] Why Are Conditional Generative Models Better Than Unconditional Ones? arXiv preprint arXiv:2212.00362.
>
> [5] Visual Chain-of-Thought Diffusion Models. arXiv preprint arXiv:2303.16187.

---

### Official Review · Reviewer_jbcg · 2023-07-06

**Soundness:** 3 good
**Presentation:** 3 good
**Contribution:** 3 good
**Rating:** 6
**Confidence:** 4

**Summary:**

The authors propose a method of fine-tuning Stable Diffusion to modify words in images, while maintaining the original font style and the background region.
Specifically, they first fine-tune the VAE with text images from several datasets.
Then, utilizing an off-the-shelf OCR detector, they randomly mask out one text box, and tune the denoising network asking it to fill the region with original text that is given as the condition in cross-attention layers.
With this simple and intuitive method, they achieve improved performance in various evaluations.

**Strengths:**

- the authors tackle a meaningful task
- the proposed method is simple and easy to reproduce
- the proposed method demonstrates improved performance on various evaluation metrics

**Weaknesses:**

- would only work for texts that can be detected by off-the-shelf OCR detectors
- there are missing details on some parts of the method

**Questions:**

- not sure whether this method is self-supervised or not, since it requires the utilization of an OCR detector in training
- could you elaborate more on a glyph image? for example, what is the output format of the glyph encoder?
- could you compute the accuracy of ChatGLM’s predictions? and how much cost does it take to fine-tune it?
- it seems that the figure 3 is redundant. maybe you can incorporate it into the figure 2?

**Limitations:**

please refer to the Weaknesses section

---

> ### Author Rebuttal · Authors · 2023-08-04
>
> We thank the reviewer for the valuable feedback and detailed review. We are encouraged that the reviewer find that our DiffUTE 'is simple and easy to reproduce' and 'demonstrates improved performance on various evaluation metrics'. We would like to response as below to adress your remaining concerns.
>
> > [W1] would only work for texts that can be detected by off-the-shelf OCR detectors
>
> As DiffUTE relies on the text boxes extracted by OCR, the generated results will not be very good when the boxes are inaccurate. Here we provide an analysis of some failed examples in this global response PDF (Figure S1-S3). However, when the OCR box can cover the text well, DiffUTE can usually perform editing work well. Our next step is to design a dynamic mask to compensate for the problem of inaccurate mask regions. We also tried an OCR-free method, fine-tuned SD1 and SD2 by instruction tuning. However, the experimental results (Table 1 and Figure 4) show that controlling text editing through instructions cannot achieve good results. This may be because natural language can only provide the semantics of the text to be modified, while glyphs can provide more detailed spatial structural information.
>
> > [W2] there are missing details on some parts of the method.
>
> Considering the limited space, if there are any unclear parts, we will provide detailed explanations in the appendix. In addition, we have provided a detailed description of the experiment's details and our training and inference code in the appendix.
>
> > [Q1] Not sure whether this method is self-supervised or not, since it requires the utilization of an OCR detector in training.
>
> Nowadays, OCR technology is quite mature and can be used for preprocessing in many downstream tasks. In fact, we only provide the position information of the text to the model, without any additional artificially defined information or labels. At the same time, OCR information is a basic requirement for scene text editing. Therefore, compared with artificially constructing an instruct-tuning dataset, our model can be regarded as a self-supervised task.
>
> > [Q2] Could you elaborate more on a glyph image? for example, what is the output format of the glyph encoder?
>
> The glyph image is created by using a universal font to write the text to be edited on a white background image. We have submitted the complete code in the support material. For easy understanding, here is a snippet of the code used to generate the glyph image. And we used trocr as the glyph encoder. Specifically, trocr consists of an encoder and a decoder. The encoder encodes the text in the image, and the decoder interprets the encoding to obtain the recognition result. In our model, we only use the encoder of trocr to extract the encoding features. Therefore, the output of the glyph encoder is a deep feature vector.
> ```python
> def draw_text(im_shape, text):
>     # text size
>     font_size = 40
>     # text font, you can download it from the internet
>     font_file = 'arialuni.ttf'
>     # create a pure white background
>     img = Image.new('RGB', ((len_text+2)*font_size, 60), color='white')
>     # define the font object
>     font = ImageFont.truetype(font_file, font_size)
>     # define the text and position
>     pos = (40, 10)
>     # write the text on background
>     draw = ImageDraw.Draw(img)
>     draw.text(pos, text, font=font, fill='black')
>     img = np.array(img)
>
>     return img
> ```
>
>
> > [Q3] Could you compute the accuracy of ChatGLM’s predictions? and how much cost does it take to fine-tune it?
>
> When ChatGLM is not fine-tuned, it is difficult to return correct results and understand human commands due to the lack of learned structured data such as OCR results. We tested it on 100 commands, and the accuracy before fine-tuning was 5%, while after fine-tuning, it reached 98%. We used an A100 for fine-tuning.
>
>
> > [Q4] it seems that the figure 3 is redundant. maybe you can incorporate it into the figure 2?
>
> Thank you for your professional comments. We will update our images in the revised version.

---

> > ### Comment · Reviewer_jbcg · 2023-08-20
> >
> > Thank you for answering my questions. It did resolve some of my concerns, and I would like to keep my original score.

---

### Official Review · Reviewer_sxrD · 2023-07-06

**Soundness:** 3 good
**Presentation:** 4 excellent
**Contribution:** 3 good
**Rating:** 6
**Confidence:** 4

**Summary:**

The paper proposes DiffUTE for general text editing.

DiffUTE utilizes Stable Diffusion model with several specific model designs, progressive training strategy, positional and glyph guidance, and a self-supervised training framework.

Equipped with these designs, DiffUTE achieves remarkable results compared to other baselines on several public datasets.
Moreover, the authors also provide a chat-based interface which enables an easier manipulation for the users.

**Strengths:**

Originality, motivation and significance:
- The paper shades an interesting perspective to edit text using pre-trained Stable Diffusion model. Two motivations raised in Line 29 and Line 32 are intuitive.
- The interaction module is interesting and easy to use.

Technical approach:
- Finetuning VAE with a progressive training strategy  (PTT) with different image sizes in different stages is a good choice to overcome blurry outputs. As shown in Table 2, with PTT, DiffUTE has a noticeable gain.
- The insight into generating fine-grained texts makes sense. With positional and glyph guidance, DiffUTE generates texts with natural shapes.
- Proposed self-supervised training strategy is straight-forward and useful. It also reduces the need of human annotations.

Clarity: the paper offers a smooth writing and is easy to follow.

**Weaknesses:**

- The motivation of using diffusion models v.s. GANs is not clearly stated. Why would the authors prefer to use diffusion model (e.g., Stable Diffusion)?
- The paper lacks some failure case analysis. For example, DiffUTE relies on pretrained OCR detector. What if the OCR detector compromises in some cases?

**Questions:**

Writing:
- Line 129: it is better explicitly to explain what $x_m$ is at the first place.


**Limitations:**

The authors emit some ethical discussions in the paper. For example, the authors should discuss the misusage of the technique for misinformation spread.

---

> ### Author Rebuttal · Authors · 2023-08-04
>
> We thank the reviewer for the valuable feedback and detailed review. We are encouraged that the reviewer find that our DiffUTE 'shades an interesting perspective to edit text using pre-trained Stable Diffusion model' and 'interaction module is interesting and easy to use'. We would like to response as below to adress your remaining concerns.
>
> > [W1] The motivation of using diffusion models v.s. GANs is not clearly stated. Why would the authors prefer to use diffusion model (e.g., Stable Diffusion)?
>
> Previously, most text editing work used GAN or simple CNN as network structures, which focused mostly on low-resolution, simple background English images and performed poorly in editing texts while maintaining text style and background consistency. Recently, diffusion models have achieved remarkable results, performing well in style preservation and detail texture. We conducted a feasibility analysis of zero-shot text editing based on GAN models and diffusion models and found that the diffusion model has great potential and can almost restore the shape of simple text in some editing tasks. Therefore, we chose the diffusion model because of its powerful generation ability as well as its controllability and scalability.
>
> > [W2] The paper lacks some failure case analysis. For example, DiffUTE relies on pretrained OCR detector. What if the OCR detector compromises in some cases?
>
> Thank you for your professional comments. Here we provide an analysis of some failed examples. Since DiffUTE relies on OCR-extracted boxes for text editing, there is no way to edit the text when the box is inaccurate, as shown in global response PDF (Figure S1-S3). However, when the OCR box can cover the text well, DiffUTE can usually perform editing work well. DiffUTE itself may also fail to generate text accurately in some scenarios. For example, when there are too many Chinese characters to be edited (more than 6), it is difficult to generate them accurately due to the complexity of Chinese character generation. Some studies that focus on font style generation only compare the performance of generating individual characters. In the future, we will consider how to edit long text sequences.
>
> > [Q1] Line 129: it is better explicitly to explain what x_m is at the first place.
>
> We would like to clarify that we have stated in line 83: "by the concatenation of latent image vector $z_t$, masked image latent vector $x_m$, and text mask $m$.", and also visualized the corresponding image in Figure 3.
>
> > [L1] The authors emit some ethical discussions in the paper. For example, the authors should discuss the misusage of the technique for misinformation spread.
>
> First, our model aims to improve users' efficiency and accuracy in image processing, especially when editing large amounts of text quickly and with high quality. Our goal is to provide users with better tools, not to spread false information. Secondly, we recognize that any technology has the risk of being misused, including our model. We will discuss this issue in our paper and propose some suggestions to minimize this risk. For example, we may recommend that regulatory agencies and social media platforms take measures to identify and combat false information, as well as provide necessary education and training to the public. At the same time, using the data we generate, we can also help improve the detection performance of false information detection models. In fact, we are also committed to developing an AI-generated content detection model for regulating generated content. Finally, we hope that users and society can be aware of the potential risks of this technology and exercise responsibility and caution when using it. We hope that our technology can bring more benefits to society rather than harm.

---

> > ### Comment · Reviewer_sxrD · 2023-08-20
> >
> > I appreciate your answers to my questions. After reading your rebuttal, I prefer to keep my original rating. Please try to include your failure case discussion and ethical discussions in the revised paper.

---

### Official Review · Reviewer_8YaX · 2023-07-06

**Soundness:** 3 good
**Presentation:** 4 excellent
**Contribution:** 3 good
**Rating:** 6
**Confidence:** 4

**Summary:**

This paper introduces DiffUTE, an innovative diffusion-based text editing framework designed to seamlessly fill in missing words in an image with user-specified text. By employing a self-supervised training framework, the model effectively learns from an extensive collection of synthetic data pairs, enabling it to infer accurate text styles and generate images that seamlessly incorporate the desired text. Experimental results showcase remarkable qualitative text editing performance from both the model's precision in both text and style accuracy. Additionally, quantitative analysis shows that the proposed method surpasses the performance of baseline approaches.

**Strengths:**

+ The proposed method exhibits impressive editing performance, as demonstrated through extensive experiments. It displays a strong ability to accurately infer text styles and generate corresponding images.

+ Leveraging LLM, the model offers broad applicability across many possible application scenarios.

+ The paper is organized clearly and is easy to read.

**Weaknesses:**

- Quantitative metrics for style: Although the paper effectively showcases the model's ability to generate text that is stylistically consistent with the rest of the images, it does not provide a quantitative analysis or specific metrics to support this claim.
- Alternative diffusion-based baselines: While ControlNet is a powerful diffusion-based editing framework, it is not specifically designed for text editing tasks. Another recent diffusion-based editing approach, DiffSTE[1], shares similarities with this work as it focuses on specialized text editing and exhibits commendable performance. How does the proposed method compare to DiffSTE in terms of performance?

[1] Improving Diffusion Models for Scene Text Editing with Dual Encoders.

**Questions:**

1. Are there any metrics available for assessing the accuracy/consistency of text style? Or potentially this can be validate through a human study similar to the Cor metrics present in paper.
2. Performance comparison with other diffusion-based text-editing method, DiffSTE?
3. An intriguing aspect of this work is its remarkable ability to accurately infer text styles, even when multiple possible texts are present within an image. For instance, in Figure 4, column 1, we can observe that the imprinted time exhibits the correct style (black), despite the presence of additional red texts. It raises the question of how robust this achievement is. Specifically, does the model consistently succeed when confronted with multiple text categories? Furthermore, can this success be attributed to different types of characters, such as numerical values versus Chinese characters, as depicted in this example?

**Limitations:**

This paper clearly discusses the limitations.

---

> ### Author Rebuttal · Authors · 2023-08-04
>
> We thank the reviewer for the valuable feedback and detailed review. We are encouraged that the reviewer find that our DiffUTE 'exhibits impressive editing performance, as demonstrated through extensive experiments' and 'leveraging LLM, the model offers broad applicability across many possible application scenario'. We would like to response as below to adress your remaining concerns.
>
> > [W1&Q1] Quantitative metrics for style: Although the paper effectively showcases the model's ability to generate text that is stylistically consistent with the rest of the images, it does not provide a quantitative analysis or specific metrics to support this claim. & Are there any metrics available for assessing the accuracy/consistency of text style? Or potentially this can be validate through a human study similar to the Cor metrics present in paper.
>
> It is difficult to quantify the effect of the style directly, but we can provide the results of a user study. Specifically, we randomly select 100 images from our Web dataset. Given each image, we can obtain 4 edited results including 3 baselines and our method. We invited 50 users to identify the edited text style in each group that they felt was most similar to the original image.
> Finally 20,000 comparison results are collected, followed by using the Bradley-Terry (B-T) model [1] to calculate an overall ranking of all methods. As presented in the following Table, our DiffUTE achieves the highest B-T score.
>
> | Method | B-T Score |
> | :---: | :---: |
> | SRNet | 0.1140 |
> | SD2-FT | 0.1545 |
> | DiffSTE | 0.3378 |
> | DiffUTE | 0.3937 |
>
> > [W2&Q2] Alternative diffusion-based baselines: While ControlNet is a powerful diffusion-based editing framework, it is not specifically designed for text editing tasks. Another recent diffusion-based editing approach, DiffSTE[1], shares similarities with this work as it focuses on specialized text editing and exhibits commendable performance. & How does the proposed method compare to DiffSTE in terms of performance? Performance comparison with other diffusion-based text-editing method, DiffSTE?
>
> Thank you for your professional comments. We have made a detailed comparison with DiffSTE on the validation set. As shown in the table below, our method performs better than DiffSTE on all datasets, which may be due to the use of instructions to control image editing in DiffSTE. Obviously, glyph-based control conditions can provide more spatial information. In addition, DiffSTE only supports English text editing and does not perform well on more difficult Chinese text editing. (The Web dataset contains text in various languages, mainly English and Chinese.)
>
> | Model | Avg.-OCR | Avg.-Cor |Web-OCR | Web-Cor | ArT-OCR | ArT-Cor | TextOCR-OCR | TextOCR-Cor | ICDAR13-OCR | ICDAR13-Cor |
> | :---: | :---: | :---: | :---: | :---: | :---: | :---: | :---: | :---: | :---: | :---: |
> | DiffSTE | 74.30 | 75 | 48.55 | 50 | 82.72 | 84 | 84.85 | 85 | 81.48 | 81 |
> | DiffUTE | **85.41 (+11.11)** | **85.5 (+10.5)**| **84.83 (+36.28)** | **85 (+35)** | **85.98 (+3.26)** | **87 (+3)** | **87.32 (+2.47)** | **88 (+3)** | **83.49 (+2.01)** | **82 (+1)** |
>
> > [Q3] An intriguing aspect of this work is its remarkable ability to accurately infer text styles, even when multiple possible texts are present within an image. For instance, in Figure 4, column 1, we can observe that the imprinted time exhibits the correct style (black), despite the presence of additional red texts. It raises the question of how robust this achievement is. Specifically, does the model consistently succeed when confronted with multiple text categories? Furthermore, can this success be attributed to different types of characters, such as numerical values versus Chinese characters, as depicted in this example?
>
> This is indeed an interesting question. In order to further understand the generation ability of DiffUTE, we have provided more examples in this global response PDF. As shown in the figure S4, when the image is filled with a single Chinese character, DiffUTE can also infer the text style based on the surrounding text. Furthermore, as shown in the figure S6 , the target we input for modification is "13", but DiffUTE generated a result with the addition of "元" based on its understanding of surrounding information. This demonstrates that DiffUTE has a certain degree of document understanding ability, and can infer the required text style based on contextual information in document data. This reasoning and learning ability can also be observed from the image S4. DiffUTE infers the angle of the text to be filled based on the posture of the surrounding text, so that the inclination angle of the text is consistent with other relevant fonts around it. We believe that the reasoning ability of DiffUTE comes from the training on a large amount of data, from which it learns some structured information. Of course, it is not always able to accurately infer and there may be situations where the style does not match the expectation, but there is no perfect model after all.
>
>
> References
>
> [1] A comparative study for single image blind deblurring. In CVPR, 2016.

---

> ### Comment · Reviewer_8YaX · 2023-08-19
> **Thank you for your replies**
>
> Thank you for the detailed replies. The additional information clarifies my previous concerns:
>
> [A1 for previous W1,Q1]: The subjective study quantitatively shows that DiffUTE can synthesize characters with good style consistency. I agree that it's difficult to quantify the text editing performance using existing metrics. Therefore, I second with reviewer FkEt that it would be great to also include the commonly used metrics, so that readers can understand the performance from different perspectives uncovered by different metrics. It is glad to see DiffUTE also performs good on these metrics based on your reply to reviewer FkEt.
>
> [A2 for previous W2,Q2]: The experiment shows DiffUTE outperforms STOA diffusion-based text editing framework.
>
> [A3 for previous Q3]: The attached examples hint that DiffUTE has the ability to infer style information (angle, font) from the context, even in a more challenging scenario when numbers and text need to be predicted simultaneously.
>
> I appreciate updates made to the paper. From my perspective, while the latent diffusion has been shown effective in many generation tasks, generating scene text is still kinds of difficult, especially in the publicly available diffusion models (e.g. stable diffusion). DiffSTE indeed shows good text editing performance. Therefore I would like to maintain my current score.

---

### Author Rebuttal · Authors · 2023-08-04

We thank the reviewers for the positive reviews and constructive feedback. We thank the AC, SAC and PC for facilitating the review process.

It is very encouraging to hear from the reviewers that:

- Performance of DiffUTE: “exhibits impressive editing performance; has strong ability to accurately infer text styles and generate corresponding images; demonstrates improved performance on various evaluation metrics; ” [8YaX, jbcg]

- Usability of DiffUTE: "Leveraging LLM, the model offers broad applicability across many possible application scenarios; the interaction module is interesting and easy to use; the incorporation of LLM into the inference process is a compelling and intriguing approach; " [8YaX, sxrD, Rgqn]

- Motivation of this work: "tackle a meaningful task; addressed  a realistic problem that current diffusion models struggle to handle effectively” [jbcg, Rgqn]

- Paper writing: "writing is clear and the motivations seem sound; paper is organized clearly and is easy to read" [FkEt, 8YaX]


We provide clarifications to each of the queries from the reviewers as response to each of the reviews. We sincerely hope that our DiffUTE would be positively received, considering its value addition to the community. We provide in the PDF attachment examples that the reviewers requested to be supplemented.

---

### Decision · Program_Chairs · 2023-09-21

**Decision:**

Accept (poster)

**Comment:**

This paper presents DiffUTE, a diffusion-based scene-text editing framework. Leveraging several proposed modules on top of the Stable Diffusion, DiffUTE is able to modify the words in images while preserving the original font style. Reviewers agree that the proposed model designs are intuitive and effective. Furthermore, reviewers also agree that DiffUTE shows remarkable performance compared with prior baselines in terms of content generation and style preservation. In the rebuttal phase, the authors actively respond to the reviewers and address the raised concerns including comparison with more baselines and metrics.